# Measuring Activities of Daily Living in Stroke Patients with Motion Machine Learning Algorithms: A Pilot Study

**DOI:** 10.3390/ijerph18041634

**Published:** 2021-02-09

**Authors:** Pin-Wei Chen, Nathan A. Baune, Igor Zwir, Jiayu Wang, Victoria Swamidass, Alex W.K. Wong

**Affiliations:** 1PlatformSTL, St. Louis, MO 63110, USA; benny.pinwei.chen@gmail.com (P.-W.C.); nabaune@wustl.edu (N.A.B.); victoria@platformstl.com (V.S.); 2Program in Occupational Therapy, Washington University School of Medicine, St. Louis, MO 63108, USA; 3Department of Psychiatry, Washington University School of Medicine, St. Louis, MO 63110, USA; zwir@wustl.edu (I.Z.); wangjiayu@wustl.edu (J.W.); 4Department of Computer Science and Artificial Intelligence, University of Granada, 18010 Granada, Spain; 5Department of Neurology, Washington University School of Medicine, St. Louis, MO 63110, USA; 6Center for Rehabilitation Outcomes Research, Shirley Ryan AbilityLab, Chicago, IL 60611, USA

**Keywords:** activities of daily living, stroke, rehabilitation, telemedicine, remote sensing technology, machine learning, wearable electronic devices

## Abstract

Measuring activities of daily living (ADLs) using wearable technologies may offer higher precision and granularity than the current clinical assessments for patients after stroke. This study aimed to develop and determine the accuracy of detecting different ADLs using machine-learning (ML) algorithms and wearable sensors. Eleven post-stroke patients participated in this pilot study at an ADL Simulation Lab across two study visits. We collected blocks of repeated activity (“atomic” activity) performance data to train our ML algorithms during one visit. We evaluated our ML algorithms using independent semi-naturalistic activity data collected at a separate session. We tested Decision Tree, Random Forest, Support Vector Machine (SVM), and eXtreme Gradient Boosting (XGBoost) for model development. XGBoost was the best classification model. We achieved 82% accuracy based on ten ADL tasks. With a model including seven tasks, accuracy improved to 90%. ADL tasks included chopping food, vacuuming, sweeping, spreading jam or butter, folding laundry, eating, brushing teeth, taking off/putting on a shirt, wiping a cupboard, and buttoning a shirt. Results provide preliminary evidence that ADL functioning can be predicted with adequate accuracy using wearable sensors and ML. The use of external validation (independent training and testing data sets) and semi-naturalistic testing data is a major strength of the study and a step closer to the long-term goal of ADL monitoring in real-world settings. Further investigation is needed to improve the ADL prediction accuracy, increase the number of tasks monitored, and test the model outside of a laboratory setting.

## 1. Introduction

Stroke is a leading cause of long-term disability, with nearly 800,000 adults in the U.S. experiencing a stroke annually [1]. Impairments in sensorimotor function, as are common following stroke, negatively impact performance in activities of daily living (ADLs) [2]. Almost half of the stroke survivors experience limitations in ADLs [3]. These limitations are a key concern among clinicians and patients. Research has shown strong relationships between performance in ADLs and patients’ quality of life [4] and the risk of re-hospitalization [5,6] after stroke. ADLs are complex sensorimotor activities involving dynamic spatial-temporal coordination of our limbs and trunk. Developing effective tools for monitoring ADLs and complex bodily movement following stroke could provide a wealth of clinically relevant information useful for tailoring therapy post-stroke.

However, clinical assessment of ADLs in stroke is limited to self-reported or clinician-rated scales, such as the Functional Independence Measure or the Barthel Index. While these measures have been clinically validated, they are mostly retrospective and susceptible to reporting bias and error [7,8,9]. An earlier study involved elderly patients and their families or nurses completing the Lawton Personal Self Maintenance scale and the Instrumental ADL (IADL) scale. They found that patients consistently rated themselves less disabled on both measures than other raters [10]. Another study found similar results in which patients reported less disability in ADL scales than family raters. The trend was held when comparing patient self-reports to researcher observations [11]. These findings indicated that patients might have trouble noticing their disability or underplay their disability. Furthermore, because these measures are rated on an ordinal scale, they may lack the resolution to detect subtle changes, limiting the ability to monitor a patient’s recovery precisely. Due to patient burden, costs, or contextual factors, clinicians infrequently implement clinical measures, so this approach is inadequate for characterizing the day-to-day variability of an individual’s function [12].

Fortunately, the last decade has seen massive growth in the capacity to collect an array of physiologic data for monitoring the health and functioning of an individual via sensors, such as those commonly found in smartphones or wearable devices [13]. Although wearable technologies have enabled automatic and continuous measurement of metrics useful for predictive medicine, such as energy expenditure or pedometers, current utilization of wearable data provides an incomplete depiction of an individual’s performance of ADLs, especially for aging and disabled populations. Mobility is often limited in these populations, reducing the utility of step-counters, heart rate measures, and other crude metrics to accurately measure an individual’s capacity to participate in meaningful ADLs. Moreover, metrics derived from wearable sensors (e.g., acceleration or angular velocity) do not have direct clinical meaning and are, therefore, difficult to use for clinical decision making. It is becoming imperative to develop better technology that provides an objective and clinically-valued measurement of an individual’s ADLs.

Human activity recognition has garnered intense research interest over the past few decades and can be generally separated into inertia-based detection and video-based detection approaches. On the one hand, video-based activity detection has been growing tremendously and has shown promising results (please refer to these reviews for details [14,15,16]). However, video-based activity detection raises privacy concerns due to potential breach of data or data misuse [17]. On the other hand, inertia-based activity recognition is less intrusive, and has shown significant improvements over time [17]. Inertia-based activity recognition systems utilize IMU’s (inertial measurement units consisting of an accelerometer and gyroscope) to measure body kinematics directly. These direct measures may provide additional information for recognizing post-stroke activities. This technology would allow researchers and clinicians to monitor rehabilitation progress, validate treatment efficacy, and detect functional decline early. Such radical change in measurement may further heighten our ability to track real-world outcomes in rehabilitation and provide accurate measures for decentralized (fully remote) clinical trials in the future.

Although the inertia-based activity recognition approach has received increasing attention [18,19,20,21], prior research was based on a large repository database of able-bodied individuals. In addition, previous research did not directly record mobility-impaired individuals (e.g., individuals who experienced a stroke) under a clinical setting for machine learning (ML) purposes. In addition, most studies have performed internal validation, where a portion of the same data set is used to test a model created from the remaining data. These are barriers to the eventual application of ADL monitoring in real-world settings. Therefore, this study aimed to develop a novel prediction model based on ML algorithms and to determine the accuracy of detecting different ADLs performed by stroke survivors using wearable sensors. We conducted this study in a simulation living room and kitchen. Lastly, we collected independent training and testing data to perform external validation, which more closely imitates real-world prediction conditions.

## 2. Materials and Methods

### 2.1. Participants

Participants were community-dwelling adults with stroke. We recruited 11 stroke survivors from the stroke registry at Washington University School of Medicine, a database of individuals who consented to future research participation at the time of their stroke hospitalization. Inclusion criteria included: (1) age 18+; (2) English-speaking; and (3) mild stroke as defined by baseline National Institutes of Health Stroke Scale (NIHSS) score from 0 to 5. We chose to study patients with mild stroke because motor function and ability to accomplish basic ADLs are only minimally affected, increasing the chance that the protocol will be fully completed in an appropriate time and allowing us to adhere to the project budget. Exclusion criteria: (1) previous neurologic or neuropsychiatric disorder (e.g., dementia, schizophrenia) that makes interpretation of the self-rated scales difficult; (2) Short Blessed Test score > 8 (indicating significant cognitive impairment); (3) history of moderate disability prior to stroke (pre-morbid Rankin Scale score < 3); (4) vision that is poorer than 20/100 (as determined by the Lighthouse Near Visual Acuity Test); (5) Apraxia screen of Tulia < 9; and (6) evidence of severe aphasia (NIHSS aphasia item >2). Eligible participants participated in our study at an ADL simulation lab in the Program in Occupational Therapy across two study visits. All participants were at least six months post their stroke incident, meaning their functional recovery had largely plateaued.

### 2.2. Procedures

During each visit, we fitted participants with five inertial measurement units (IMUs) to collect accelerometer and gyroscope data: one on each wrist, one on each of their upper arms, and one on their hip. The IMU used in this study were Apple Watch Series 3 (Apple Inc., Cupertino, CA, USA). We developed the system using the Apple Watches due to their affordability, commercial availability, and ability to transfer motion data wirelessly over users WiFi network. We utilized two study visits to collect independent training (visit two) and testing (visit one) data sets. During visit one, we used a method similar to that introduced by Bao and Intille [18] for capturing naturalistic behaviors. Participants engaged in a series of ADL tasks following a standardized script for the examiner to provide minimal guidance. The examiner informed participants what tasks they would perform next but did not comment on how to achieve them. For example, the examiner would ask participants to cook stir-fry following a recipe (which would require gathering ingredients, chopping, cooking), then to eat a serving (which would require plating the food, retrieving eating utensils, and sitting down to eat). Participants were allowed to shift between activities naturally and perform actions like chopping and cooking as they preferred. The experimenter aimed to provide as little guidance as possible, giving cues for the next steps and answering questions; otherwise, participants operated primarily independently. This care was taken to most closely mimic real-world scenarios of ADL prediction. Participants performed these semi-naturalistic activities to record testing data for evaluating the ML algorithms. We used surveillance cameras to record and label all activities from visit one and later review for possible errors.

During visit two, participants performed four minutes of each atomic activity (a total of 19 atomic activities; Table 1). An atomic activity is a simple movement involved in ADL tasks (e.g., stirring a pan or chopping vegetables are two atomic activities involved in cooking). Atomic activities require participants to continuously perform specific movements with experimenter guidance, which contrasts with semi-naturalistic activities where we provided participants with a high-level goal (e.g., cook pancakes) but did not give explicit performance instructions. Participants performed atomic activities repeatedly for four minutes; for example, the participant would stir pancake batter for four minutes, place/retrieve spices on/from a shelf for four minutes, or fold laundry for four minutes. We collected atomic activity performance data to train our ML algorithms. The experimenter recorded the timing of atomic activity performance using in-house software (henceforth, activity labeling software) on a seventh-generation iPad. While four minutes of atomic activity data were collected per activity, participants could perform activities in multiple blocks to reduce boredom or fatigue due to the homogenous repeated motion. Participants performed the semi-naturalistic activities first (visit one) before they completed the atomic activities (visit two) to reduce possible bias. Specifically, we avoided participants performing atomic activities first because the homogenous repeated motion might influence the way participants performed the semi-naturalistic activities.

Prior to this study, we tested on healthy adults and achieved adequate accuracy when collecting only two minutes of atomic activity data per activity. Nevertheless, we collected four minutes per atomic activity in this study to increase the amount of training data. We anticipated this amount of data to be beneficial due to potential variability between post-stroke participants. We could ensure adequate data remained for each activity in any case that segments of data had to be removed due to errors (e.g., technical difficulties with IMU recording). The four-minute was chosen as the upper-limit as a precaution to avoid participant over-exertion. We could further ensure data were collected for all activities within the allotted three-hour study sessions. The study sessions were all facilitated by a licensed occupational therapist trained to monitor these activities and ensure patient safety.

We chose the 19 activities (Table 1) based on several criteria. We determined the final ADL and IADL list through consultation with therapists and physicians specialized in stroke. Activities chosen were: (1) commonly used to assess post-stroke functional activity, (2) amenable to performance in our simulated living environment, and (3) feasible for mild stroke patients to perform during the allotted three-hour testing sessions. This study received ethics approval from the institutional review board at Washington University. All participants provided written consent and received an honorarium to acknowledge their research contribution.

### 2.3. Data Analysis

We used descriptive statistics to characterize the study sample. We combined initial data from five devices for each participant and assigned activity labels.

These predictor variables included the translational and rotational acceleration along three axes (x, y, z) across each of the five sensors, resulting in 30 variables (Table A1 of Appendix A). We used three-second epochs to summarize the raw data into feature space with time-domain (e.g., mean, standard deviation, autocorrelation, and slope) and frequency-domain features (see Figure A1 of Appendix A for one variable as an example). This technique provided stationarity for the time series, allowing each activity to be analyzed as a stationary stochastic process in model development [22]. Data were preprocessed, including slicing data with an optimal window size to avoid redundancies, input variables were normalized and then compressed into 6012 data points. We chose three-second epochs based on our previous testing in healthy adults, where three-second was the optimal epoch length while testing five epoch lengths (integers 1 through 5).

Motion data from the semi-naturalistic activity session (visit 1) was labeled using the data from our activity labeling software. We used data from visit 2 (atomic activities) for the training, whereas data from visit 1 (semi-naturalistic activities) for the testing. We implemented t-Distributed Stochastic Neighbor Embedding (t-SNE) analysis and cluster analysis to analyze the samples’ distribution in terms of the first and second visits.

The classes/tasks’ sizes for activities substantially vary among the observed subjects, so we rebalanced the training data sets to prevent over-teaching the classifier only to predict the major (negative) classes. We accomplished this using both the Synthetic Minority Oversampling Technique (SMOTE), a bootstrapping algorithm that provides more data points for the smaller class based on variable distributions of that class [23,24,25] and the Extended Nearest Neighbor (ENN) algorithm that downs size the larger class [25] for the training data. It should be noted that both SMOTE and ENN were only used on the training data after the data were split into the training and validation data sets. Moreover, since learning the hyperparameters involved the 70–30% hold-out validation, SMOTE and ENN was used only in 70% hold-out. We also used the Shapley Additive exPlanations (SHAP) to identify the contribution of the individual IMU features in the model predictions [26,27].

### 2.4. Classified Model Development

We achieved the primary classification model with the gradient boosting model (GBM), XGBoost [28]. This method has been shown to have improved supervised classifier model formalization, compared to other GBMs or random forest algorithms, and better efficiency and controls for overfitting [29].

Hyperparameters supported by the XGBoost package were fine-tuned using the Tree Parzen Estimator (TPE) Bayesian optimization algorithm [30] to find the best combination of parameters based on the 70–30% hold-out validation. The fitness loss function was defined by the AUC [29]. Data were randomly partitioned into two sets based on the hold-out sampling partition used for the internal training validation test with unbalanced classes of atomic activities.

There was no need for other sampling methods rather than hold-out because the external semi-naturalistic sample was completely disjoint, including variability in terms of times of tasks, order, completeness, and the presence of the training tasks. The external independent validation dataset based on semi-naturalistic activities was not seen at any training session or data processing. Tasks were selected by following a standard stepwise regression using XGBoost, where models were fitted based on the choice of predictive variables carried out by an automatic procedure. In each step, a variable is considered for addition to or subtraction from the set of explanatory variables based on some pre-specified criterion. The stop iteration criteria was a performance of 90% ± 2%.

All scripts were written in python, and the XGBoost package and the other classifiers were used with a python sklearn API (https://scikit-learn.org (accessed on 9 December 2020)) [28,31]. For parameter fine-tuning, we used the hyperopt package (https://scikit-learn.org/stable/modules/grid_search.html (accessed on 9 December 2020)). We used the imbalance package to apply SMOTE and ENN. We used Sklearn for all data partitions and the calculation of the metrics.

We selected the final set of features, as estimated above, and then tested multiple ML models: Decision Tree, Random Forest, SVM, and XGBoost to determine the optimal classification model. Default parameters for each of these methods were learned as described above (sklearn-scikit) by optimizing the AUC metric. We assessed model performance using accuracy, recall (sensitivity), precision (positive predictive value), and ROC (receiver operating characteristic) curve (AUC) metrics [32]. The loss function was based on AUC.

## 3. Results

### 3.1. Characteristics of Study Participants

Table 2 provides an overview of participant demographic characteristics. The majority of participants were males (72%), right-handed (91%), and experienced an ischemic stroke (100%). The average age was 60 years old. 55% (*n* = 6) participants experienced a right hemispheric stroke, 36% (*n* = 4) experienced a left hemisphere stroke, and one participant did not have information on the side of stroke. The average time since the stroke incident was 2.76 years (SD = 1.73).

### 3.2. Data Extraction and Model Development

The total duration of the observed data was 153,159 s. We divided data into three-second epochs; thus, the total epochs were 51,053. It should be noted that we encountered a hardware malfunction for one participant’s visit, but we did not identify this malfunction until the experiment was completed. As a result, we excluded a vast amount of data of this participant for subsequent analyses. For feature selection, we first utilized the t-SNE graph to understand whether there were dissociations between training data from visit two and testing data from visit one (Figure 1). We found no significant separations between the two data segmentation, suggesting no dissociations between the two datasets. This result indicates that we can safely use the training set based on atomic measurements in visit 2 for predicting semi-naturalistic activities in visit 1.

Figure 2 shows three confusion matrices corresponding to classifications based on 19, 10, and 7 tasks. We selected tasks retained in the model based on the stepwise regression process, in which we removed one ADL/IADL task on each run with the lowest accuracy. Using the XGBoost algorithm, we achieved an average accuracy of 97% on the training set and an accuracy of 90% in an independent test based on seven tasks in a sample composed of all subjects while performing semi-naturalistic activities. We also obtained 0.91 for recall (sensitivity), 0.83 for precision, and 0.98 for AUC in the independent test set. The seven tasks include cutting, vacuuming, sweeping, spreading jam or butter, folding laundry, eating, and brushing teeth. Ten tasks produced 86% accuracy on the training set and an 82% accuracy on the independent test set. Ten tasks include the previous seven tasks plus: Taking off/putting on a shirt, wiping cupboards, and buttoning a shirt. We further computed the t-SNE distributions of the ten tasks and found many of them clustered together (Figure 3), suggesting that these are predictable structured classes. SHAP indicated that, among the five IMUs, both wrists and the hip captured the most critical information for activity recognition. The important features contained gyroscopic standard deviation and accelerometer standard deviation and mean. Table 3 shows the performance evaluation metrics across ML models. We found that XGBoost optimizes or equates with methods that produce the best results. These results provided initial evidence that ADL/IADL functioning can be predicted with adequate accuracy by using wearable sensors (IMUs) and ML algorithms. We also used the SHAP values to identify the contribution of the individual IMU features to predict activities in the model prediction (Figure 4). Among 5 IMUs, both wrists and the hip, capturing the most critical information for predicting activities. Essential IMU features included standard deviation and mean features from gyroscope and accelerometer.

## 4. Discussion

Using motion-based data collected from IMUs, we have developed and validated ML algorithms to recognize a list of ADLs among a sample of community-dwelling stroke survivors. A significant contribution of our study is the use of independent training and testing data sets. We asked participants to complete semi-naturalistic activities in an obstacle course design (e.g., unload groceries and then prepare and eat stir-fry using ingredients). We also asked them to complete atomic activities recorded within a block design (e.g., four minutes of chopping vegetables). After that, we trained the ML algorithms with the atomic activities and tested them with semi-naturalistic activities. We found that our ML algorithms could recognize a set of activities in a semi-naturalistic environment with adequate accuracy. Enabling participation in daily activities is the ultimate outcome in rehabilitation. These results provide initial evidence that motion-based sensors and ML, especially the XGBoost and SVM algorithms, can predict post-stroke daily activities.

Among all 19 activities included in the ML model, we found the top three activities that can be accurately classified are sweeping (0.95), eating with hands (0.88), and vacuuming (0.84), whereas other activities that had the lowest accuracy are serving on a plate (0.25) and pan stirring (0.32). We have inspected the raw data and found that these high accuracy activities often involve highly repetitive movements that can be tracked with frequency-domain features. In contrast, the low accuracy activities consist of high variations in the data (i.e., movements are less repetitive or occasionally occur within a 3 s epoch). For example, serving on a plate is an activity that often involves multiple movements that happen in a short time and the motion sequence is not highly repetitive. Participants would grab the spatula, raise the food item, and then place the food item on a plate occurring approximately less than 10 s. The motion sequence of these steps is not repetitive during the semi-naturalistic recordings. Pan stirring is an activity in which participants would stir the food once and stop for a while before the next stirring, making the model hard to distinguish these motions within the 3 s epoch window.

Human activity recognition is a growing field. A prior review summarized all accelerometer-based human activity recognition research and found that past research focuses primarily on mobility activities (e.g., sitting and standing) or postures [33]. Another review paper summarized all applications of activity recognition systems. It found that ML algorithms utilizing video recordings (e.g., Infra-red and RGB-D) have tremendous improvement in recognizing the types of activities [17]. Further review of video-based detection systems can be found in these papers, summarizing the current status of the research [14,15,16]. Although video-based systems are useful for security or surveillance solutions, it is not as valuable when utilizing it for patient monitoring due to privacy concerns.

Human activity recognition using IMUs or other wearable sensors has remained a likely alternative to video-based systems. Most wearable sensors allow measurement of an individual’s physiological signals (e.g., heart rate) and direct kinematics (e.g., intensity and moderate-to-vigorous physical activity). These measures may provide additional information for recognizing activities among older adults and individuals with physical disabilities, such as those after the stroke. However, there are not many accelerometry-based ML research that directly records mobility impaired individuals. A study involving a group of stroke survivors utilized wearable smartphones to predict six “activity” labels: Sit, stand, lie, stairs, large movements (including walking, small steps, and opening doors), and small movements. This research was able to achieve high accuracy in predicting these “activities” [34]. Another study evaluated the likelihood of recognizing postures of individuals with a stroke while performing activities using smartphones. This research found that increasing the complexity of activities decreased model accuracy in detecting postures [35]. These two studies have demonstrated the potential of using mobile technologies and ML methods for human activity recognition.

Many of the previous studies utilized a database from able-bodied individuals [19,20,21]. Models built using data from able-bodied individuals may not generalize to stroke survivors. Indeed, models built solely from mild stroke patient data may not generalize to moderate or severe stroke. The generalizability of models is a major concern in building clinically valid models. The current study was designed as a pilot study to present the feasibility of our approach to ADL/IADL monitoring. We restricted this study to mild stroke patients to ensure they could perform all activities and in the allotted study session times. We acknowledge such limitations and in future studies we will need to address these concerns. For example, it may be possible that a single model across mild, moderate, and severe stroke survivors is feasible or perhaps different groups require unique models. An additional common shortcoming in prior research is the reliance on internal validation to assess the accuracy of models. An example of internal validation includes building a model using one portion of the data set and testing it with the remaining portion. This approach is likely to reduce the ecological and clinical relevance of a model. Using clean data (no interruptions or spurious data) to train the model is important, through predicting activity using clean data may also exaggerate the model utility. Naturalistic performance is not clean. For example, individuals often transition between activities frequently, pausing to talk and communicating with their hands while doing so. A major contribution of our study is the use of clean data to train our models and an independent semi-naturalistic dataset to validate the models. The goal was to match real-world performance as closely as possible for our testing dataset. We acknowledge that the simulated living environment and the structured script to guide the activities are not a perfect model of natural activity. However, we took many precautions to better match natural performance conditions; the high degree of accuracy we achieved using data from stroke patients and through external validation using an independent dataset are major strengths of our study. In future studies we plan to transition into performance in the user home to match real-world performance even more closely.

Along with other activity recognition studies, our current study added to the literature by harnessing motion sensors and building ML algorithms to detect different types of at-home activities and directly collecting activity data in stroke survivors. Our research provides initial evidence to predict (i.e., recognize) ADL functioning in stroke. More importantly, the methodology and technology developed to collect and analyze data in this study may translate into inpatient rehabilitation facilities for other populations because these facilities often have ADL simulation rooms.

During the experiment, we did not observe any particular activity that participants had significant difficulty completing. This is likely due to our participants, who are mild stroke survivors with minimal motor and ADL problems. Nevertheless, we observed that some participants demonstrated slight difficulties with certain bilateral activities. We reviewed the raw data and found certain activities have a higher variation of inertial intensity. Specifically, spreading butter/jam and eating with hands have the highest variations of inertial intensity, suggesting that these two tasks demand more irregular motions performed by the upper limbs. On the other hand, wiping the cupboard or pan stirring has a lower variation, suggesting these tasks require less fine-motor ability. Although differentiation of difficulty level of various activities is not the scope of our current study, our current ML model could classify some of them with high accuracy.

Furthermore, to explore our technology’s clinical utility in characterizing patients’ ADL performance, we have conducted a supplementary analysis of two selected activities that typically require bilateral use of the upper limbs (Figure A2 of Appendix B). This analysis aimed to explore the degree of usage of upper limbs while performing specific ADL tasks. According to prior research [36,37], bilateral limb usage impairment is common for individuals after a stroke. To quantify the real-world activity, an accelerometry-based outcome can be used to examine each upper limb’s intensity of usage while a patient performs a bilateral task [38]. This outcome measure further helps clinicians and researchers quantify the difficulty level of various activities. For example, clinicians may use the plots seen in Figure A2 to understand the dominant and non-dominant limbs’ usage demand across multiple activities. As seen, when compared to the “spreading butter/jam, “chopping vegetables” demanded higher intensity on the dominant hand relative to the non-dominant hand. Knowing the usage demand of various activities can help develop the treatment plan by selecting appropriate ADL tasks for interventions that match the patient’s motor function. Other possible outcomes are the amount of time spent on specific activities and the repetition and variation among those activities.

There are several limitations to this study. First, we have a small sample size for model building. As this is a pilot study, the current research provides the first step in designing a better paradigm for large-scale data collection. We suspect that it is essential to collect high-quality movement data from the target population. Thus, we will continue to hone our data collection techniques. For example, we will further improve our customized activity labeling software, which allows us to label movement data into activity categories and reduce the number of resources required to conduct ML research, a major barrier in activity recognition research. Second, we collected activity data in a semi-naturalistic environment instead of participants’ homes. This methodology may reduce the generalizability of detecting activities in real-world settings. Nevertheless, this study was designed to test the data collection method and provide initial proof-of-concept to detect daily activities from individuals with a stroke. The semi-naturalistic environment serves as an intermediate step to gain knowledge and experience to record data before testing it in participants’ homes. Further investigation is also needed to improve the ADL prediction accuracy, increase the number of tasks monitored, and test the model outside of a laboratory setting.

Developing more capable models and verifying their efficacy in a larger and more diverse sample of stroke patients will be a critical next step, although these initial results are highly promising. We will work closely with clinicians and patients as we move forward. We will collaborate to determine the most appropriate activities to monitor and the most valuable ways to present the findings. We understand this user-centered approach will be of utmost importance if we hope for our technology to solve some of the most critical issues plaguing rehabilitation practice and research. If successful, the advancements could benefit stroke survivors and a wide range of individuals with disabilities. Remote monitoring of activities crucial to functional independence may bring clinicians and researchers much closer to talent the challenging goal of individualized/personalized medicine.

## 5. Conclusions

Enabling patients’ daily activity and participation in life is the ultimate goal for rehabilitation. Current results in this pilot study have shown the possibility to monitor task-specific ADLs from IMU data collected in a simulated setting and modeled with the ML methodology. With the ADL data reflecting patients’ real-world functional performance, clinicians and researchers can have insight into whether the treatment positively impacts an individual’s everyday life.

## Figures and Tables

**Figure 1 ijerph-18-01634-f001:**
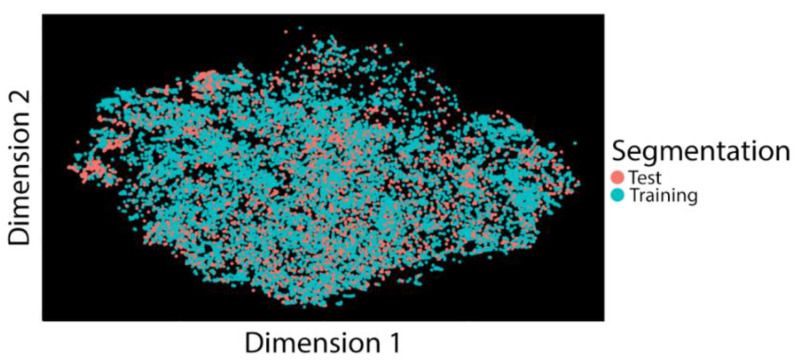
t-Distributed Stochastic Neighbor Embedding (t-SNE) graph showing segmentation for both training and test data sets.

**Figure 2 ijerph-18-01634-f002:**
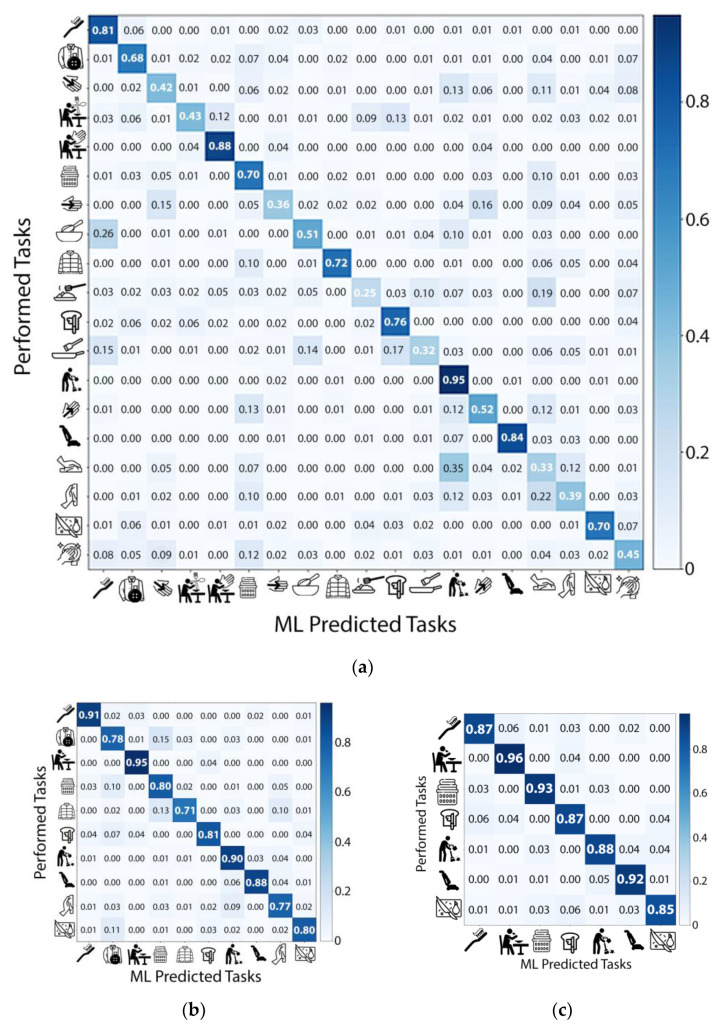
Confusion matrices corresponding to classifications based on (**a**) 19 tasks, (**b**) 10 tasks, and (**c**) 7 tasks. Tasks selection was performed following a standard stepwise regression process using XGBoost.

**Figure 3 ijerph-18-01634-f003:**
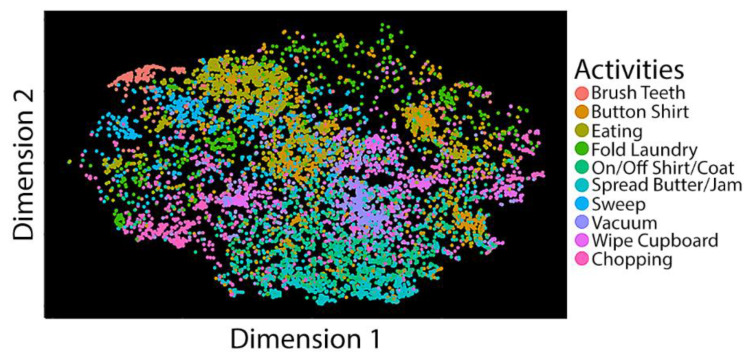
t-SNE graph for the ten ADL tasks. Dots indicate data from different ADL tasks.

**Figure 4 ijerph-18-01634-f004:**
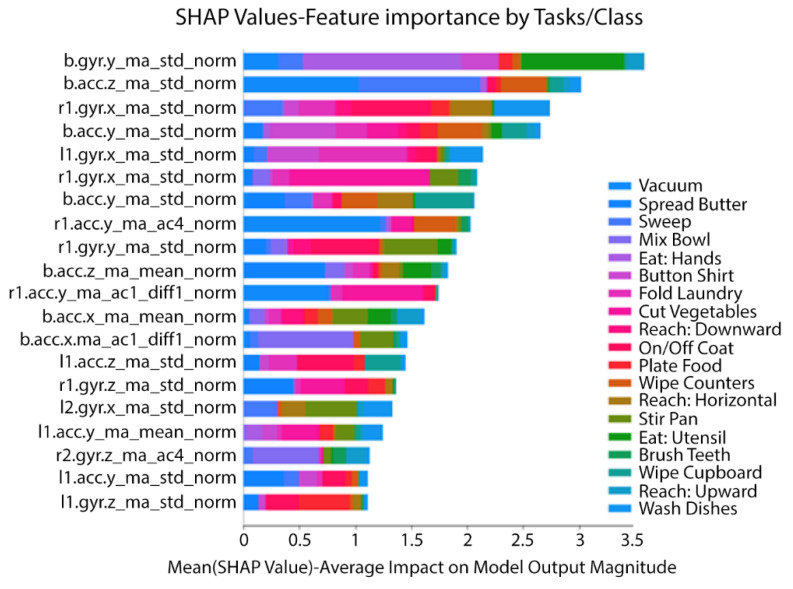
Influence of various inertial measurement units (IMU) features for predicting ADL tasks on the XGBoost model.

**Table 1 ijerph-18-01634-t001:** Atomic activities with categories label between activities of daily living (ADL) or instrumental activities of daily living (IADL).

Atomic Activities	ADL/IADL
Brush Teeth	ADL
Mixing Powders	IADL
Spreading Butter/Jam	IADL
Eating with Hands	ADL
Eating with Utensils	ADL
Buttoning shirt/coat	ADL
Put and take off the coat	ADL
Moving items horizontally	ADL
Reaching Up	ADL
Reaching Down	ADL
Washing Dishes	IADL
Chopping	IADL
Pan Stirring	IADL
Serve on a plate	IADL
Sweeping	IADL
Vacuuming	IADL
Wiping horizontal surface	IADL
Wiping vertical surface	IADL
Folding clothes	IADL

**Table 2 ijerph-18-01634-t002:** Demographic characteristics of the participants (*n* = 11).

Characteristics	Mean (SD)/Count
Age	59.64 (9.04)
Years since stroke	2.76 (1.73)
NIHSS ^1^ Total Score	1.45 (1.29)
Sex	8 males; 3 females
Race	6 Caucasian; 4 African American; 1 unknown
Education	4 high school graduates; 5 some college; 2 bachelor’s degree
Handedness	10 right; 1 left
Stroke Type	11 ischemic; 0 hemorrhagic
Stroke Side	6 right-hemispheric; 4 left-hemispheric; 1 unknown

^1^ NIHSS = National Institutes of Health Stroke Scale.

**Table 3 ijerph-18-01634-t003:** Performance metrics of seven ADL tasks classification across machine learning (ML) models.

Performance Metric ^1^	Decision Tree	Random Forest	SVM	XGBoost
Training Set				
Accuracy	0.56	0.79	0.97	0.97
AUC	0.74	0.88	0.99	0.98
Precision	0.50	0.80	0.97	0.97
Recall	0.56	0.79	0.97	0.97
Test Set				
Accuracy	0.43	0.80	**0.90**	**0.90**
AUC	0.68	0.89	0.95	**0.98**
Precision	0.47	0.84	**0.92**	0.83
Recall	0.43	0.80	0.90	**0.91**

^1^ Note: Numbers in bold represent the best values across ML models.

## Data Availability

Data is upon requested with authors’ approval.

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
