# Peer review of "Measuring Activities of Daily Living in Stroke Patients with Motion Machine Learning Algorithms: A Pilot Study"

_ijerph, 2021, doi:10.3390/ijerph18041634_

Round 1
Reviewer 1 Report
This study presented the use of wearable devices and machine learning for activity detection in individuals with mild stroke. While the study is an important addition to the existing literature, some aspects of the study are not clearly presented. Please find detailed comments below:
- The introduction is missing information regarding existing literature. There are existing studies with similar aims that have not been cited or discussed.
- The make, model, and manufacture information for the IMU sensors used is missing and should be added in the methods section.
- Was there no inclusion/exclusion criteria related to time since injury for the study? How do you think this affects their ability to perform the tasks in the study?
- Details of the methods are not clear in parts- did participants repeat activities for 4 minutes during visit-2, or did they hold a static position (e.g.: reaching up) for 4-minutes? What is the rationale for performing each activity for 4-minutes during visit-2?
- What is the rationale for choosing the specific activities performed in this study?
- What is the rationale for choosing 3-second epochs?
- Details of the ML algorithms in the study are missing- What parameters were tuned for each model (Decision tree, random forest, SVM and XGBoost)? What are the values of the parameters for each optional classifier model (Decision tree, random forest, SVM and XGBoost)?
- The ML models were evaluated on quite a few performance metrics (Accuracy, AUC, Precision, etc.)- which metrics did you use to choose the 'best' model?
- To provide more clarity in results, please present the results of ML models for all activities before presenting results for 7 activities where the model had greater accuracy.
- The discussion should include a paragraph regarding the choice of the activities performed in the study and their relationship to the prediction accuracy. When the ML models were tested on all activities, which activity had the top misclassification rate? Why do you think that happened? Did some patients have a hard time performing certain activities well? What was the level of difficulty of each activity for all patients?
- One major limitation of the study is that it included individuals with mild stroke only, and therefore the generalizability of study results is limited. Please add a paragraph with relevant information in the discussion.
Author Response
This study presented the use of wearable devices and machine learning for activity detection in individuals with mild stroke. While the study is an important addition to the existing literature, some aspects of the study are not clearly presented. Please find detailed comments below:
- The introduction is missing information regarding existing literature. There are existing studies with similar aims that have not been cited or discussed.
In the introduction section, we added two paragraphs to introduce the missing existing literature by citing review papers that summarized the current status of the work (lines 85-109). Furthermore, we added a paragraph in the discussion section to identify the research gap and support our study’s findings (lines 365-391).
- The make, model, and manufacture information for the IMU sensors used is missing and should be added in the methods section.
The IMU’s sensors were all Apple Watch Series 3. This was added to the methods section (lines 137-140).
- Was there no inclusion/exclusion criteria related to time since injury for the study? How do you think this affects their ability to perform the tasks in the study?
Thanks for the comment. Our study participants were at least 6 months post their stroke incident. Specifically, the average time since our study participant’s stroke incident was 2.76 years (SD=1.73). We believe that their functional recovery had largely plateaued. This information was added in lines 132-133 and 264-265).
- Details of the methods are not clear in parts- did participants repeat activities for 4 minutes during visit-2, or did they hold a static position (e.g.: reaching up) for 4-minutes? What is the rationale for performing each activity for 4-minutes during visit-2?
We added further clarification to the procedures section (lines 162-173 and lines 174-183). During visit two, participants performed four minutes of each atomic activity. While recording an atomic activity, participants repeatedly performed that activity without stopping (e.g., stirring pancake batter with a whisk or folding laundry repeatedly). The cumulative time recorded per activity was tallied by the iPad app so that the four minutes of activity could be performed in multiple discrete blocks to avoid fatigue, boredom and to provide breaks if needed. For instance, we always split brushing teeth into a minimum of two separate blocks, since it could become uncomfortable. In contrast, most participants could vacuum for four minutes straight since it is a fairly low effort activity. Furthermore, in our previous tests in healthy adults (unpublished feasibility testing), we had achieved high accuracy using 2 minutes of atomic activity data to train our ML models. We increased this to four minutes to anticipate that stroke patient performance will be less homogeneous than healthy adults and account for potentially lost data. Generally, more training data is better; we settled on four minutes as our upper limit to fit all activities within the testing sessions.
- What is the rationale for choosing the specific activities performed in this study?
We added a paragraph in the procedures section to provide the rationale (lines 184-188).
- What is the rationale for choosing 3-second epochs?
We added a paragraph in the data analysis section to provide the rationale (lines 205-207).
- Details of the ML algorithms in the study are missing- What parameters were tuned for each model (Decision tree, random forest, SVM and XGBoost)? What are the values of the parameters for each optional classifier model (Decision tree, random forest, SVM and XGBoost)?
We updated the main text and included the exact site where the default parameters of each method were optimized (lines 246-254). All scripts were written in python, and the XGBoost package and the other classifiers were used with a python sklearn API (https://scikit-learn.org). For parameter fine-tuning, we used the hyperopt package (https://scikit-learn.org/stable/modules/grid_search.html).
- The ML models were evaluated on quite a few performance metrics (Accuracy, AUC, Precision, etc.)- which metrics did you use to choose the 'best' model?
The loss function was based on AUC as stated in the main text (lines 256-257). Also, we bolded the best values of performance metrics in Table 3 to indicate the best model.
- To provide more clarity in results, please present the results of ML models for all activities before presenting results for 7 activities where the model had greater accuracy.
We added the results of ML models for all 19 tasks (Figure 2a).
- The discussion should include a paragraph regarding the choice of the activities performed in the study and their relationship to the prediction accuracy. When the ML models were tested on all activities, which activity had the top misclassification rate? Why do you think that happened? Did some patients have a hard time performing certain activities well? What was the level of difficulty of each activity for all patients?
Thanks for the thoughtful comments. We provided clarification about how we chose the activities (lines 241-245). Furthermore, we added a paragraph to discuss the high versus the low classified activities and provided a justification, based on our observation from the data to explain the cause of these misclassifications (lines 327-340). We also added Figure 4 to visualize the importance of each feature per activity.
- One major limitation of the study is that it included individuals with mild stroke only, and therefore the generalizability of study results is limited. Please add a paragraph with relevant information in the discussion.
We added further elaboration to the discussion section (lines 365-374).
Reviewer 2 Report
This paper focuses on the usage of machine learning algorithms and sensors to detect specific activities of daily living for persons having experienced a stroke.
The approach is centered on the detection of simple activities such as serving a plate or sweeping. By using motion sensors, raw data is fed to ML algorithms, most notably XGBoost but also others (SVM etc) in order to classify the activity.
Although the subject of this paper is interesting and relevant, given the need to assess the capabilities of patients having suffered a stroke, there are a number of issues that need to be discussed.
First, and as acknowledged by the authors, the number of participants used to test the overall approach is very low. With only eleven participants it is not possible to take conclusive results. At best, it help motivating further and more comprehensive studies that may validate the approach in the future. This does not mean that the presented study is useless. All the contrary, it does provide some promising results, but this should be reflected in the title, abstract and the rest of the manuscript. The auhtors may require to substantially change the scope of the paper, in order ot make it clear that it is a pilot study and that the results are only an early stage indication.
Second, although in the introduction there is a compelling discussion about the importance of ADL assessment for stroke patients, the actual technical content of the paper focuses on identification of these activities. This is not enough to effectively characterize the level of proficiency or ability of the person to perform these tasks. This is still an important next step that is not yet on the scope of this paper.
Third, the conditions of the study are still too strictly scripted and guided by the experiment settings. There is evidently a problem of bias and potential distance with real life conditions. These factors cannot be overlooked and further diminish the significance of the experiments results.
The paper is generally well written and is easy to follow.
Author Response
This paper focuses on the usage of machine learning algorithms and sensors to detect specific activities of daily living for persons having experienced a stroke.
The approach is centered on the detection of simple activities such as serving a plate or sweeping. By using motion sensors, raw data is fed to ML algorithms, most notably XGBoost but also others (SVM etc) in order to classify the activity.
Although the subject of this paper is interesting and relevant, given the need to assess the capabilities of patients having suffered a stroke, there are a number of issues that need to be discussed.
First, and as acknowledged by the authors, the number of participants used to test the overall approach is very low. With only eleven participants it is not possible to take conclusive results. At best, it help motivating further and more comprehensive studies that may validate the approach in the future. This does not mean that the presented study is useless. All the contrary, it does provide some promising results, but this should be reflected in the title, abstract and the rest of the manuscript. The auhtors may require to substantially change the scope of the paper, in order ot make it clear that it is a pilot study and that the results are only an early stage indication.
This is a great point. We made changes throughout the title and body text to frame this article as a pilot study and to better discuss the shortcomings and benefits of the study. Notably, we addressed the issue of generalizability in the context of our fairly homogenous patient sample in the discussion section (lines 365-391).
Second, although in the introduction there is a compelling discussion about the importance of ADL assessment for stroke patients, the actual technical content of the paper focuses on identification of these activities. This is not enough to effectively characterize the level of proficiency or ability of the person to perform these tasks. This is still an important next step that is not yet on the scope of this paper.
We addressed this in part by better referencing past work. Existing studies often limit ML prediction to very broad categories of activity (e.g., standing, sitting, laying down) or have had limited success in predicting ADLs. This study aimed to provide initial evidence of a high accuracy approach to predicting ADL tasks. By presenting this paper as a pilot study and providing the reasoning for limiting the study to mild stroke, we hope that readers would understand that this current study is the first step for our ADL monitoring. Furthermore, we proposed some next steps for future studies in the discussion section (lines 365-391, and lines 439-441).
Although we did not have data to expand on the analyses that could better understand individual performance, we attempted to discuss how we could use the data in the future to characterize usage between limbs (frequency of use and intensity per activity). Laterality could give a clear indication of disuse as well as changes in use and how that relates to others. With a much larger and more diverse sample, we intend to investigate other outcomes as well, such as movement smoothness and eventually prognosis (lines 339-425).
Third, the conditions of the study are still too strictly scripted and guided by the experiment settings. There is evidently a problem of bias and potential distance with real life conditions. These factors cannot be overlooked and further diminish the significance of the experiments results.
Our concern over bias was that the “atomic” (repeated) activity performance would influence the semi-naturalistic performance, which is why semi-naturalistic performance was always recorded at the first session. We elaborated in the discussion on the common use of internal validation (one data set for training and testing) in most prior studies and how our use of an independent semi-naturalistic testing data set is a major improvement. While we tried to match real-world performance the best we could, we acknowledge that a lab will never be a perfect replacement for the home. Our future plans are to transition into testing in patients' homes without any prompting (see lines 170-173 and lines 375-391).
The paper is generally well written and is easy to follow.
Thank you.
Round 2
Reviewer 1 Report
The authors have addressed most comments and concerns well. Although the introduction could use a little more polishing, the manuscript adds value to the existing literature related to activity recognition in individuals post stroke.
Author Response
The authors have addressed most comments and concerns well. Although the introduction could use a little more polishing, the manuscript adds value to the existing literature related to activity recognition in individuals post stroke.
We streamlined the texts in the introduction section (lines 43-115).
Reviewer 2 Report
Thanks to the authors for their answers and for the work done to address previous comments to the original manuscript.
In my opinion, all the major problems have been addressed. Although the main criticisms remain (very small number of subjects, and lab testing limitations wrt. real tests), the current version of the paper justifies and explains why this contribution is still valuable. Moreover, it is explicit about the preliminary status of some of these results, while describing the future directions and how they will gain from the outcome of this paper.
Minor details:
Lines & comments:
91: On the other hand -> missing 'on the one hand' phrase
93 -> systemS
Fig 4 is not very readable
368: moderate or stroke either -> ??
Author Response
Thanks to the authors for their answers and for the work done to address previous comments to the original manuscript.
In my opinion, all the major problems have been addressed. Although the main criticisms remain (very small number of subjects, and lab testing limitations wrt. real tests), the current version of the paper justifies and explains why this contribution is still valuable. Moreover, it is explicit about the preliminary status of some of these results, while describing the future directions and how they will gain from the outcome of this paper.
Thank you. We much appreciate that you have found values in the paper, although this study has several limitations.
Minor details:
Lines & comments:
91: On the other hand -> missing 'on the one hand' phrase
We added “On the one hand” accordingly (lines 87-93).
93 -> systemS
We made the change as suggested (line 93).
Fig 4 is not very readable
We updated Figure 4 with better resolution.
368: moderate or stroke either -> ??
We changed the text to indicate models built solely from mild stroke patient data may not generalize to moderate or severe stroke (lines 369-370).